



**A 16-year dataset (2000-2015) of high-resolution (3 hour, 10 km) global surface**
**solar radiation**
Wenjun Tang[1,2], Kun Yang[3,2], Jun Qin[1], Xin Li[1,2], and Xiaolei Niu[1]
1. National Tibetan Plateau Data Center, Institute of Tibetan Plateau Research,
Chinese Academy of Sciences, Beijing 100101, China.
2. CAS Center for Excellence in Tibetan Plateau Earth Sciences, Chinese Academy of
Sciences, Beijing 100101, China.
3. Ministry of Education Key Laboratory for Earth System Modeling, Department of
Earth System Science, Tsinghua University, Beijing 100084, China.
Corresponding author and address:
Dr. Wenjun Tang
Institute of Tibetan Plateau Research, Chinese Academy of Sciences
Building 3, Courtyard 16, Lin Cui Road, Chaoyang District, Beijing 100101, China
Email: tangwj@itpcas.ac.cn
Tel:    +86-10-84097046
Fax:    +86-10-84097079



**Abstract:** The recent release of the International Satellite Cloud Climatology Project
(ISCCP) HXG cloud products and new ERA5 reanalysis data enabled us to produce a
global surface solar radiation (SSR) dataset: a 16-year (2000-2015) high-resolution (3
h, 10 km) global SSR dataset with an improved physical parameterization scheme.
The main inputs were cloud optical depth from ISCCP-HXG cloud products, the
water vapor, surface pressure and ozone from ERA5 reanalysis data, and albedo and
aerosol from Moderate Resolution Imaging Spectroradiometer (MODIS) products.
The estimated SSR data was evaluated against surface observations measured at 42
stations of the Baseline Surface Radiation Network (BSRN) and 90 radiation stations
of the China Meteorological Administration (CMA). Validation against the BSRN
data indicated that the mean bias error (MBE), root mean square error (RMSE) and
correlation coefficient ($R$) for the instantaneous SSR estimate at 10 km scale were
-11.5 W m$^{-2}$, 113.5 W m$^{-2}$, and 0.92, respectively. The error was clearly reduced when
the data were upscaled to 90 km; RMSE decreased to 93.4 W m$^{-2}$ and $R$ increased to
0.95. For daily SSR estimates at 90 km scale, the MBE, RMSE and $R$ at the BSRN
were -5.8 W m$^{-2}$, 33.1 W m$^{-2}$ and 0.95, respectively. These error metrics at the CMA
radiation stations were 2.1 W m$^{-2}$, 26.9 W m$^{-2}$ and 0.95, respectively. Comparisons
with other global satellite radiation products indicated that our SSR estimates were
generally better than those of the ISCCP flux dataset (ISCCP-FD), the global energy
and water cycle experiment surface radiation budget (GEWEX-SRB), and the Earth's
Radiant Energy System (CERES). Our SSR dataset will contribute to the land-surface
process simulations and the photovoltaic applications in the future. The data set is
available at https://doi.org/10.11888/Meteoro.tpdc.270112 (Tang, 2019).
**Keywords**: Surface solar radiation; Global product; High-resolution; Parameterization
scheme
## 1. Introduction


Surface solar radiation (SSR), which drives the energy, water and carbon cycles
of Earth's system, is the driving input for simulations of hydrology, ecology,
agriculture and land-surface processes (Wild, 2009; Wang et al., 2012). The accuracy
of SSR data influences simulations of runoff, gross primary productivity,
growth and yield of crops, and land data assimilation (Wild, 2012; Jia et al., 2013).
SSR is also an important variable that affects the speed of glacier melting (Yang et al.
2011). Variations of SSR also affect the rate of global warming and the change of pan
evaporation (Wild, et al., 2007; Qian et al., 2006).
Information on the spatiotemporal distribution of SSR is fundamental for
selection of sites for solar power plants, decisions on energy policy, optimization of
solar power systems, and operations managment (Mondol et al., 2008; Sengupta et al.,
2018). To address issues such as these, historical SSR data has been obtained mainly
through ground-based observations, station-based estimates, and satellite-based
retrievals (Pinker & Laszlo, 1992; Li and Leighton, 1993; Liang et al., 2006; Zhang et
al., 2004; Wang et al. 2011; Huang et al., 2011; Kato et al., 2013; Ma & Pinker, 2012;
Zhang et al., 2014; Wang et al., 2015; Niu and Pinker, 2015).
Measurement by accurately calibrated and well-maintained radiometer of
pyranometer is the most effective method to obtain reliable long-term SSR data.
Although these data are valuable for simulations of land surface processes, solar
power applications and evaluation of satellite retrievals (Sengupta et al., 2018), the
high cost of maintaining radiation radiometers means that networks of radiation
stations are too sparsely distributed. However, networks of routine meteorological
stations are denser than those of radiation stations, and the variables observed at
routine meteorological stations can be used to estimate SSR. For example, based on



sunshine duration data, Tang et al. (2013, 2018) constructed long-term datasets of
both daily global radiation and direct radiation over China at more than 2400 routine
meteorological stations of the China Meteorological Administration (CMA). These
datasets are generally more accurate than those derived from satellite retrievals (Yang
et al., 2010). However, station-based estimates of SSR can be conducted only at
routine weather stations, many of which are sparsely distributed, often in remote
regions and harsh environment.
Alternatively, remote sensing retrievals based on satellites can provide reliable
spatiotemporally continuous SSR data, either globally or regionally. The many
methods that have been developed to retrieve SSR from satellite data can be roughly
divided into three types: empirical, semi-empirical and physical.
Empirical methods build function relationships between SSR measured at limited
numbers of stations and satellite data by applying regression or artificial intelligence
technology (Lu et al., 2011; Wei et al., 2019). Empirical methods may work well at
some locations, but the ability to expand their coverage to broader regions is limited.
Semi-empirical methods generally combine a physical model for clear-sky
conditions with an empirical scheme for cloudy conditions. A well-known
semi-empirical method is the Heliosat method of Cano et al. (1986), from which
several improved versions have since been developed (Hammer et al., 2003; Mueller
et al., 2009; Posselt et al., 2012; and Wang et al., 2014).
Physical methods are generally well-suited to generalization because they take
into account the physics processes of transfer of solar radiation from the top of the
atmosphere to the Earth's surface. The look-up table (LUT) and physical
parameterization methods (Pinker & Laszlo, 1992; Liang et al., 2006; Lu et al, 2010;
Qin et al., 2015; Xie et al., 2016; Huang et al., 2018) are two typical physical methods



that were widely used to estimate SSR from satellite data.

Several well-known global SSR datasets have been produced by physical

methods. These include the global energy and water cycle experiment surface
radiation budget (GEWEX-SRB, Pinker and Laszlo, 1992), the International Satellite
Cloud Climatology Project flux dataset (ISCCP-FD, Zhang et al., 2004) and the
Earth's Radiant Energy System (CERES) radiation products (Kato et al., 2013).
Although each of these have been widely used in various fields, the spatial resolutions
(>=100 km) of these SSR products is too coarse to meet the requirements of
high-resolution SSR data. A high-resolution (5 km, 1 h) global SSR product of the
Global Land Surface Satellite (GLASS) were recently released, but it contains data
spanning only three years. Tang et al. (2016) also produced a high-resolution SSR
product by combining data from polar-orbit and geostationary satellites, but the
product covers only China and the dataset spans only eight years.

The greatest uncertainty in satellite retrievals of SSR is the lack of a high-quality

cloud product, which severely limits the development of high-resolution, long-term
global satellite SSR products. However, the release in 2017 of new, global, long-term
ISCCP H-series cloud products at a spatial resolution of about 10 km has provided an
opportunity to develop a long-term high-resolution global-scale climate dataset of
SSR.

We developed a global-scale 16-year dataset (2000-2015) of SSR data from the

new ISCCP H-series cloud products and ERA5 reanalysis data, validated the accuracy
of this dataset with surface observations, and compared its performance with other
global satellite products. Section 2 introduces the method we used to estimate SSR.
Section 3 describes the input data we used for SSR estimation and the observations
data used for SSR validation. In Section 4, we presented our evaluation of the SSR





product and compared it with other satellites products. Data availability is given in
Section 5, and Section 6 presents some conclusions and explores future work to
further improve SSR products.

**2 Estimation of SSR**
The method we used to estimate SSR with ISCCP H-series cloud data is mainly
based on the SUNFLUX scheme, which was developed by Sun et al. (2012; 2014) and
first used by Tang et al. (2017) to retrieve SSR data from Moderate Resolution
Imaging Spectroradiometer (MODIS) atmospheric and land products. Their validation
of their results against measurements at BSRN stations indicated a mean root mean
square error (RMSE) of ~90 W m$^{-2}$ for instantaneous SSR. Although Tang et al.
(2017) achieved higher accuracy than we did in this study (because the MODIS cloud
products they used are generally of better quality than the ISCCP H-series cloud data),
the instantaneous SSR they retrieved is slightly overestimated at most stations because
the original method they used only considers the effect of aerosol scattering on SSR,
but ignores the effect of aerosol absorption. To overcome this issue, we replaced the
aerosol parameterization scheme used by Tang et al. (2017) with that used by Qin et
al. (2015) and Tang et al. (2016). The resultant method is a pure physical
parameterization scheme with an efficient calculation speed. The inputs to the method
include cloud optical depth (COD) in the visible band, cloud cover, aerosol optical
depth (AOD), surface pressure, precipitable water, total ozone, surface albedo, and
carbon dioxide concentration (fixed at 375 ppm by volume). Detailed information
about the method is provided by Tang et al. (2017) and Tang et al. (2016).

**3 Data**



### 3.1 Input data


To produce the 16-years SSR products at global scale, we used three types of
input data.
The first of these was the level 2 ISCCP H-series cloud product HXG (H-series
pixel-level global, here called ISCCP-HXG), which is a globally merged product
generated based on the HGS (H-series gridded by satellite) product. The resolutions of
HXG are 3 h and 10 km, and the HXG cloud products are available for the period
from July 1983 to December 2015. Note that the ISCCP-HXG data are $0.1^{\circ}$ gridded
snapshots (or instantaneous) available every 3 h not the average value over 3 h. More
information about the ISCCP-HXG cloud product is provided by Young et al. (2018).
Four variables were used in the ISCCP- HXG cloud product: cloud mask, VIS
retrieved liquid cloud optical depth, VIS retrieved ice cloud optical depth and cloud
top temperature. The cloud mask was used to distinguish clear-sky pixels from cloudy
pixels and the cloud top temperature was used to distinguish liquid cloud and ice
cloud.
The second data type we used was the new ERA5 reanalysis data. Three
variables of the ERA5 reanalysis data were used: surface pressure, total column water
vapor and total column ozone. The resolutions of the ERA5 reanalysis data are 1 h
and 25 km. To derive the same spatial resolution as the ISCCP- HXG cloud product,
we re-sampled the three variables of ERA5 reanalysis data to a spatial resolution of 10
km.
The third data type comprised aerosol and albedo data. The MODIS aerosol
(MOD08 or MYD08) and albdeo (MCD43A3, Schaaf et al., 2002) products were used.
MOD and MYD denote product obtained from Terra and Aqua platforms,
respectively, and MCD indicates a combined product processed from both platforms



(King et al., 2003). The spatial resolution of MODIS aerosols and albedo data are
about 100 km and 5 km, respectively, so we re-sampled them both to 10 km. Missing
values in the MODIS aerosol and albedo products (included the period of 1 Jan 2000
to 23 Feb 2000) were replaced with the corresponding values of monthly mean
climatological data. Note that the use of climatological data to replace the real
information of aerosol and albedo would have introduced some uncertainty. Thus,
care should be taken when using the SSR dataset we derived for trend analysis.

**3.2 Validated data**

In this study, we used radiation observations made in 2009 to validate the

accuracy of the global-scale SSR estimate. These radiation observations were
collected at two networks. The first set was the radiation observations (with temporal
resolution of 1 minute) measured at 42 stations of Baseline Surface Radiation
Network (BSRN, Ohmura et al, 1998), which were marked as red crosses in Figure 1.
Radiation observations measured at BSRN stations are regarded as the most reliable
radiation data due to the instruments of highest available accuracy and careful
maintenance (see website: https://bsrn.awi.de/). To reduce uncertainty caused by
cosine response error of the pyranometers, we did not use the measured global
radiation data; instead we used the total of the measured direct and diffuse radiation to
evaluate the accuracy of the retrieved SSR.

The second set was the daily radiation observations measured at 90 CMA

radiation stations, which were denoted by black circles in Figure 1. Though the
pyranometers used to measure global radiation at CMA radiation stations were
calibrated by a series of standard procedures (Yang et al., 2008), the observed
radiation data collected at CMA radiation stations frequently include questionable





values, which may have been a result of improper operation of instruments and/or
instrument defects (Shi et al., 2008). To reduce the uncertainty caused by the
questionable radiation data, we used a quality-check procedure (Tang et al. 2010) to
exclude the spurious and erroneous measurements.

**4 Results and Discussion**
**4.1 Validation of estimated SSR against observations at BSRN stations**
Firstly, the estimated SSR were validated against the observations measured at
the 42 BSRN stations at both instantaneous and daily scales. To reduce the
uncertainties induced by broken clouds, we validated the estimated instantaneous SSR
against hourly mean observed ones centered on the time of satellite overpass,
according to the suggestion of Wang and Pinker (2009). To examine the effect of
different spatial resolutions on the accuracy of our SSR estimates, in addition to the
10 km spatial resolution, we also evaluated our estimated SSR at spatial resolutions of
30, 50, 70, 90 and 110 km derived by averaging the SSR values observed at the
original scale of 10 km.
Accuracy for instantaneous SSR at 90 km scale (RMSE = 93.4 W m$^{-2}$, $R$ = 0.95;
Fig. 2, Table 1) was clearly superior to that at 10 km scale (RMSE 113.5 W m$^{-2}$, $R$ =
0.92), which may indicate that the surface observation points are generally
representative of more than 10 km, especially under cloudy conditions. Another
possible reason for this phenomenon would be caused by the time mismatch between
satellite observation and surface observation because the satellites do not take
instantaneous snapshots of the entire Earth. Generally, the last generation
geostationary satellites, such as the Geostationary Operational Environmental Satellite
(GOES), take about 30 min to scan the entire Earth. The averaging inherent in



upscaling of spatial resolution would tend to decrease these time mismatches.

To further illustrate this issue, the performances of our instantaneous SSR with

different spatial resolutions at the 42 BSRN stations were given in Table 1, which
suggests that the accuracy was clearly improved when the data were upscaled to 30
km, with a further slight improvement at 70 km, but that accuracy started to decrease
at 90 km. The performance of the ISCCP-FD was also presented in Table 1.
Apparently, the accuracy of our estimated instantaneous SSR is significantly higher
than that of the ISCCP-FD. A further advantage of our dataset is that its spatial
resolution is far higher than that of the ISCCP-FD products.

Figure 3 shows the spatial distribution of RMSE for the estimated instantaneous

SSR (spatial resolution 90 km) at all individual BSRN stations. The RMSE was < 90
W m$^{-2}$ at 30 of the 42 BSRN stations. RMSE values were between 90 and 105 W m$^{-2}$
at five stations and > 105 W m$^{-2}$ at seven stations. The 12 stations where RMSE
values were >= 90 W m$^{-2}$ are generally in coastal areas, on islands and in the
Antarctic polar region. Part of the reasons for these large error are the same as that
explained by Tang et al. (2017), who estimated instantaneous SSR with MODIS
level-2 land and atmospheric products. For example, the large RMSE value for station
IZA can be attributed to the poor representativeness of the station, which is located on
a mountain top, and this station point can not represent the satellite observations.
Another reason for the large RMSE values may be the uncertainties contained in the
inputs, especially uncertainties in cloud and aerosol data. The great uncertainties for
the MODIS AOD retrieval over coastal or island stations (Anderson et al, 2013)
would lead to large RMSE values at these stations. The large errors for the two
Antarctic stations (SYO and GVN) may reflect failure of cloud detection, which is
difficult over Antarctica region because the similarity of the properties of cloud and



surfaces snow over the Antarctica Pole, and because the temperature of cloud is
generally not lower than that of surface snow (Zhang et al. 2013).
Figure 4 presents the validation results for our estimated daily SSR at 42 BSRN
stations. The MBE values were -6.1 and 5.8 W m$^{-2}$ for spatial resolutions of 10 and 90
km, respectively. The RMSE for 10 km was 38.0 W m$^{-2}$, and its value was decreased
to 33.1 W m$^{-2}$ for 90 km. The $R$ for 10 km was 0.93 and its value was increased to
0.95 for 90 km. Table 2 also lists the performances of our daily SSR estimate with
different spatial resolutions and the performance of the ISCCP-FD daily SSR product.
Our estimates of daily SSR at all spatial resolutions were clearly more accurate than
that of ISCCP-FD, and they obviously improved when upscaled to more than 30 km.
The spatial distribution of RMSE for our estimated daily SSR at spatial
resolution of 90 km (Fig. 5) showed that RMSE at most of the 42 BSRN stations were
<35 W m$^{-2}$, although there were four stations with RMSE between 35 and 40 W m$^{-2}$
and six with RMSE >40 W m$^{-2}$. These higher RMSE values may be attributed to lack
of representativeness for some stations, errors in the inputs and uncertainty of the
algorithm, similar to the reasons for the higher errors in our estimates of instantaneous
SSR.
GWEWX-SRB and CERES are two other well-known and widely used global
satellite radiation products. Zhang et al. (2013; fig. 8) evaluated the performance of
GEWEX-SRB SSR products with the mean 3-h observed data from the BSRN and
found that RMSEs for the instantaneous and daily SSR of GEWEX-SRB were 88.3
and 35.5 W m$^{-2}$, respectively. To compare our results with those derived from
GEWEX-SRB by Zhang et al. (2013), we re-evaluated our estimated SSR with the
mean 3-h observed data from the BSRN. The RMSEs for our estimated instantaneous
and daily SSR at 10 km spatial resolution were 108.1 and 36.5 W m$^{-2}$, respectively,





both of which are greater than those of GWEX-SRB. However, when we upscaled our
estimated SSR to 90 km scale, RMSEs for our instantaneous and daily SSR were
lower, 82.4 and 30.6 W m$^{-2}$, respectively, indicating that our estimates of SSR were
more accurate than those of GEWEX-SRB at the same spatial resolution. We also
compared the performance of our estimates of SSR with that of CERES
(SYN1deg_Ed4A, Fig. 6). The accuracies of CERES were generally higher than those
of ISCCP-FD at both instantaneous and daily scales, but obviously lower than those
of our estimates at all spatial resolutions from 10 to 110 km (Fig. 6 and Table 2).
Thus, our estimated SSR based on ISCCP-HXG cloud products provided a more
accurate, higher spatial resolution dataset than those of ISCCP-FD, GEWEX-SRB and
CERES products.

**281 4.2 Validation of estimated SSR against observations at 90 CMA radiation**

**282 stations**

Our estimated SSR were further evaluated against the observations collected at
the 90 CMA radiation stations at both daily and monthly scales. Figure 7 presents the
validation results for the estimated daily SSR at spatial resolutions of 10 and 90 km.
The MBE, RMSE and $R$ for our estimated daily SSR at 10 km spatial resolution were
1.8 W m$^{-2}$, 32.4 W m$^{-2}$ and 0.93, respectively. Accuracy clearly improved for spatial
resolutions up to 90 km, for which the corresponding metrics were 2.1 W m$^{-2}$, 26.9 W
m$^{-2}$ and 0.95. The RMSE for our estimate of daily SSR at 10 km spatial resolution is
comparable to that of GEWEX-SRB daily SSR, which was also validated against
observations at the CMA radiation stations (RMSE 32. 2 W m$^{-2}$; see figure 7b of Qin
et al., 2015). However, the RMSE for the GEWEX-SRB daily SSR is clearly higher
than that of our estimate of daily SSR at 90 km spatial resolution, thus indicating that



the accuracy of our daily SSR estimates is superior to that of the GEWEX-SRB daily
SSR product at the same spatial resolution.

Table 3 shows that the accuracy of our estimates of daily SSR clearly improved

when upscaled to 30 km spatial resolution and were most accurate at 90 km spatial
resolution. RMSE and $R$ (36.5 W m$^{-2}$ and 0.91, respectively) for daily SSR of
ISCCP-FD show that our estimates are more accurate at all spatial resolutions. The
spatial distribution of RMSE for our daily SSR estimate at 90 km spatial resolution
was also given in Figure 8. Only nine CMA stations had RMSE >35 W m$^{-2}$ (Fig. 8);
most of these stations are in southern China where there is generally more cloud and
its distribution is more complicated than in other parts of China (Yu et al., 2001).

Figure 9 presents the validation results for our estimated monthly SSR. The MBE,

RMSE and $R$ for our estimated monthly SSR at 10 km spatial resolution were 1.9 W
m$^{-2}$, 16.3 W m$^{-2}$ and 0.97, and the corresponding values for 90 km were changed to
2.2 W m$^{-2}$, 14.9 W m$^{-2}$ and 0.97. It can be clearly seen that the accuracy of the
ISCCP-FD monthly SSR are inferior to our estimated monthly SSR at scales from 10
to 110 km (Table 4).

The performances for CERES daily and monthly SSR were evaluated against

observations at the 90 CMA radiation stations (Fig. 10) and also compared with
those of our estimates from ISCCP-HXG (Table 4). The MBEs for CERES daily and
monthly SSR were greater than those of our estimates at all scales, and the RMSE
for CERES daily SSR was slightly smaller than that of our estimates at 10 km spatial
resolution, but obviously greater that our estimates at spatial resolutions from 30 to
110 km. The RMSE for CERRES monthly SSR was greater than those of our
estimates at all scales. Thus, the accuracy of our estimates is generally higher than
that of CERES.




**4.3 Spatial distribution of the annual SSR**

Figure 11 presents the comparison of the global distribution of the annual mean
SSR in 2009 between our retrievals and the ISCCP-FD SSR product. From the figure,
it can be seen that the global distribution for our SSR estimate based on the ISCCP-
HXG cloud products is almost the same as that of the ISCCP-FD SSR product, but the
spatial resolution of our estimate is far higher than that of ISCCP-FD. There
is no doubt that we can get more details that the coarse resolution product ISCCP-FD
can not capture. For example, the region of high SSR clearly identified over the
Tibetan Plateau by our estimate (Fig. 11a) is barely discernible in the
ISCCP-FD-derived data (Fig. 11b). The high values are mainly from
around the equator and the low latitudes, and the low values mainly over the high
latitudes and the Arctic and Antarctic regions. This phenomenon is primarily
determined by the solar elevation angle. In addition, the relatively high values are also
found over the Bolivian Plateau, the Tibetan Plateau, and other high altitude regions
due to less radiative extinction over high altitudes.

**5 Data availability**

The 16-year dataset of global SSR is available at the National Tibetan Plateau
Data Center (https://doi.org/10.11888/Meteoro.tpdc.270112, Tang, 2019), Institute of
Tibetan Plateau Research, Chinese Academy of Sciences.

**6 Conclusions and Future work**

This study produced a 16-year (2000-2015) global dataset of SSR based on
recently updated ISCCP H-series cloud products, new ERA-5 reanalysis data and




MODIS albedo and aerosol products with a physically based retrieval scheme. The
retrieved SSR dataset was evaluated globally with observations collected at BRSN
and CMA radiation stations. To investigate the effect of spatial scale on the accuracy
of our estimated SSR dataset, our estimated SSR at spatial resolutions from 10 km to
110 km were validated. Validation against observed BSRN data showed MBEs of
−11.0 and 6.0 W m$^{-2}$ for our estimates of instantaneous and daily SSR, respectively.
RMSEs for our instantaneous and daily SSR estimates at 10 km spatial resolution
were 113.5 and 38.0 W m$^{-2}$, respectively, but their accuracy clearly improved when
upscaled to more than 30 km spatial resolution. For example, the RMSEs decreased to
93.4 and 33.1 W m$^{-2}$ when our estimates were upscaled from 10 to 90 km. The
accuracies of our SSR estimates were clearly higher than those of GEWEX-SRB,
ISCCP-FD and CERES products. At 10 km spatial resolution, validation of our daily
and monthly SSR estimates against observed data from CMA radiation stations
provided RMSEs of 32.4 and 16.3 W m$^{-2}$, respectively, but these values decreased to
26.9 and 14.9 W m$^{-2}$ when our estimates were upscaled to 90 km spatial resolution.
The errors of our SSR estimates when validated against observed data from CMA
were also lower than those of GEWEX-SRB, ISCCP-FD and CERES products. We
attributed large errors in our estimates at some stations to the lack of
representativeness of some stations, uncertainties in the input data, such as cloud
detection failures at stations in polar regions, large uncertainties in MODIS AOD
retrievals over stations in coastal areas and on islands, and uncertainty in the retrieval
scheme we used. However, the retrieval scheme we used worked well at most of the
stations used in our study despite their considerable geographic and climatologic
differences.
The spatial resolution and accuracy of the new dataset are both higher than those



of the global satellite radiation products of GEWEX-SRB, ISCCP-FD, and CERES
and will contribute to photovoltaic applications and research related to simulation of
land surface processes. When reliable global aerosol and albedo datasets become
available, we intend to expand our dataset of SSR estimates back to mid-1983. We
also plan to expand the dataset beyond 2015 by using SSR estimates from
new-generation geostationary satellites.

**Author contributions.** All authors discussed the results and contributed to the
manuscript. WT calculated the dataset, analyzed the results, and drafted the
manuscript.

**Competing interests.** The authors declare that they have no conflict of interest.

**Acknowledgments.** The CMA radiation station data were obtained from the National
Meteorological Information Center (NMIC) and. The ISCCP-HXG cloud products
were obtained from the NOAA's National Centers for Environmental Information
(NCEI). The ERA5 reanalysis data and MODIS albdedo and aerosol data were
downloaded from official websites (https://www.ecmwf.int and
https://ladsweb.modaps.eosdis.nasa.gov). The authors would like to thank the
Baseline Surface Radiation Network (BSRN) observation teams for their maintenance
work.

**Financial support.** This work was supported by the National Key Research and
Development Program of China (Grants No. 2018YFA0605400 and
2017YFA0603604), the National Natural Science Foundation of China (Grants No.



41671372), the Youth Innovation Promotion Association CAS (No. 2017100),
and the 13th Five-year Informatization Plan of Chinese Academy of Sciences
(Grant No. XXH13505-06), and the Strategic Priority Research Program of Chinese
Academy of Sciences (Grant No.XDA20100102).

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



**Figure captions**
**Figure 1** Distribution of radiation measurement stations used to evaluate the
performance of the estimated SSR. The blue circles mark the locations of
the 90 CMA radiation stations, and the red crosses mark those of the 42
BSRN stations. Note that two stations (labeled as DAR and DWN) in
Australia and two stations (labeled as BIL and E13) in America are very
close to each other.
**Figure 2** Comparisons of our estimated instantaneous SSR at spatial resolutions of (a)
10 km and (b) 90 km with observed SSR for 42 BSRN stations.
**Figure 3** Spatial distribution of RMSE (W m$^{-2}$) for our estimated instantaneous SSR
(spatial resolution 90 km) at 42 BSRN stations.
**Figure 4** Comparisons of our estimated daily SSR at spatial resolutions of (a) 10 km
and (b) 90 km with observed SSR for 42 BSRN stations.
**Figure 5** Spatial distribution of RMSE (W m$^{-2}$) for our estimated daily SSR (spatial
resolution 90 km) at 42 BSRN stations.
**Figure 6** Comparison of CERES SSR products with observed SSR at 42 BSRN
stations for both (a) instantaneous and (b) daily scales.
**Figure 7** Comparisons of our estimated daily SSR at spatial resolutions of (a) 10 km
and (b) 90 km with observed SSR at 90 CMA radiation stations.
**Figure 8** Spatial distribution of RMSE (W m$^{-2}$) for our estimated daily SSR (spatial
resolution 90 km) at 90 CMA radiation stations.
**Figure 9** Comparisons of our estimated monthly SSR at spatial resolutions of (a) 10
km and (b) 90 km with observed monthly SSR at 90 CMA radiation
stations.
**Figure 10** Comparison of CERES (a) daily and (b) monthly SSR products with those



observed at 90 CMA stations.
**Figure 11** Spatial distribution of global annual mean SSR (W m$^{-2}$) of (a) ISCCP-HXG
and (b) ISCCP-FD in 2009.

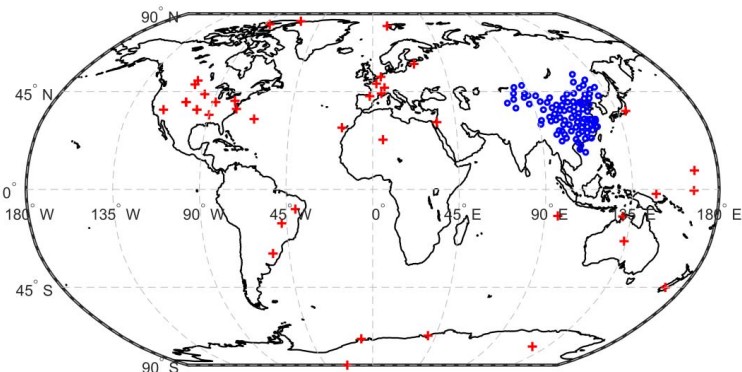


**Figure 1** Distribution of radiation measurement stations used to evaluate the performance of the estimated SSR. The blue circles mark the locations of the 90 CMA radiation stations, and the red crosses mark those of the 42 BSRN stations. Note that two stations (labeled as DAR and DWN) in Australia and two stations (labeled as BIL and E13) in America are very close to each other.



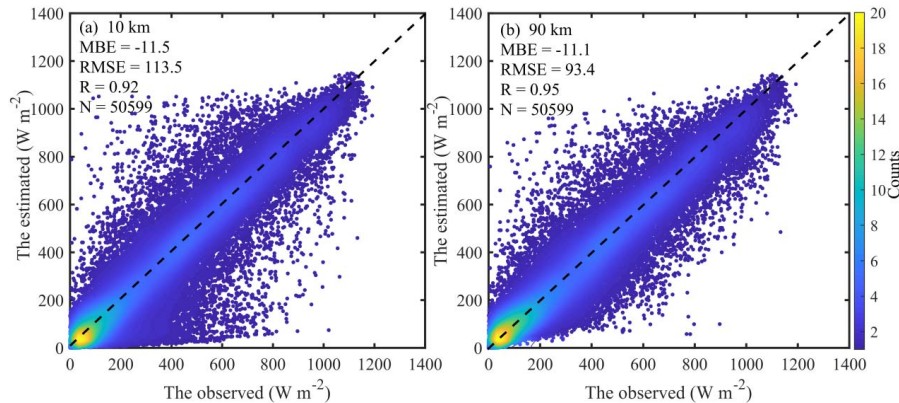

**Figure 2** Comparisons of our estimated instantaneous SSR at spatial resolutions of (a)

10 km and (b) 90 km with observed SSR for 42 BSRN stations.





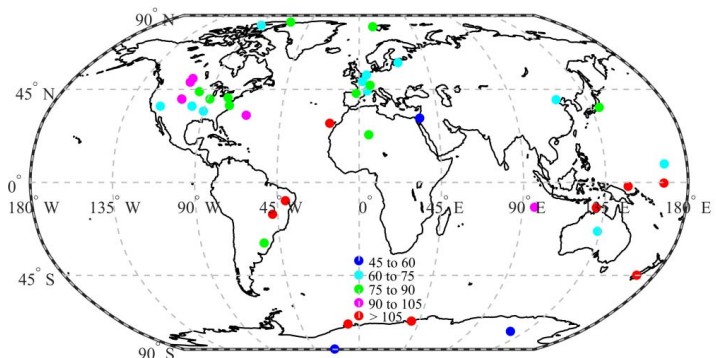

**Figure 3** **S**patial distribution of RMSE (W m$^{-2}$) for our estimated instantaneous SSR

(spatial resolution 90 km) at 42 BSRN stations.





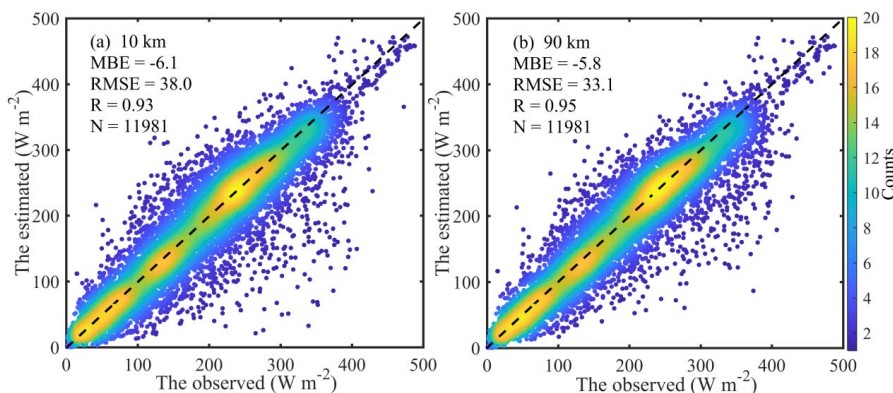

**Figure 4** Comparisons of our estimated daily SSR at spatial resolutions of (a) 10 km and (b) 90 km with observed SSR for 42 BSRN stations.





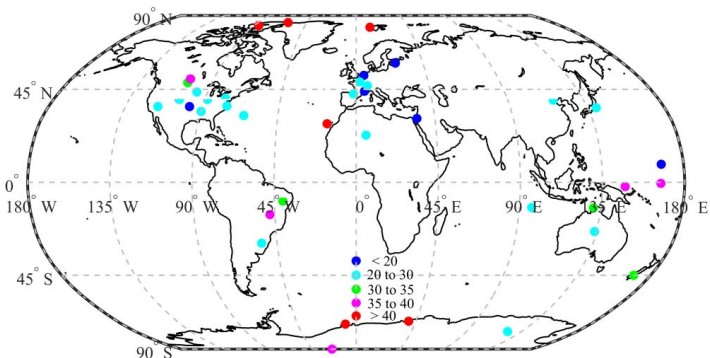

**Figure 5** Spatial distribution of RMSE (W m$^{-2}$) for our estimated daily SSR (spatial

resolution 90 km) at 42 BSRN stations.





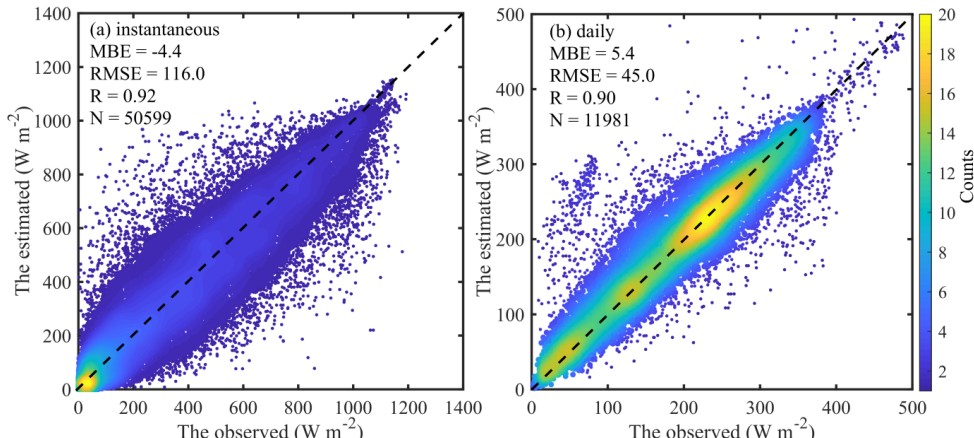

**Figure 6** Comparison of CERES SSR products with observed SSR at 42 BSRN stations for both (a) instantaneous and (b) daily scales.

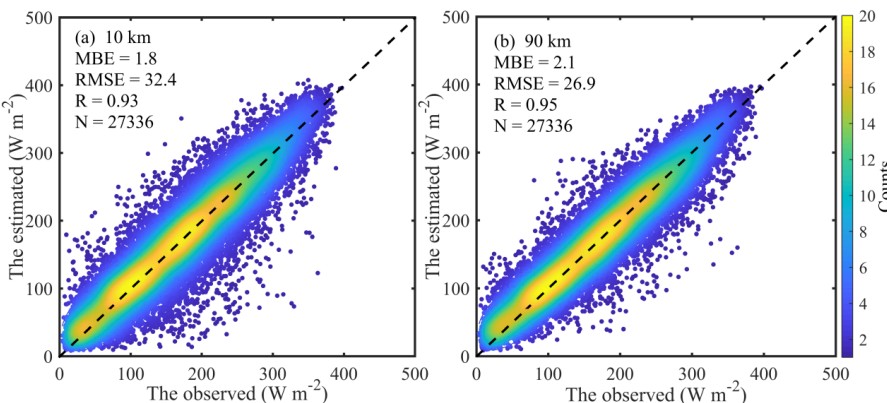

**Figure 7** Comparisons of our estimated daily SSR at spatial resolutions of (a) 10 km

and (b) 90 km with observed SSR at 90 CMA radiation stations.

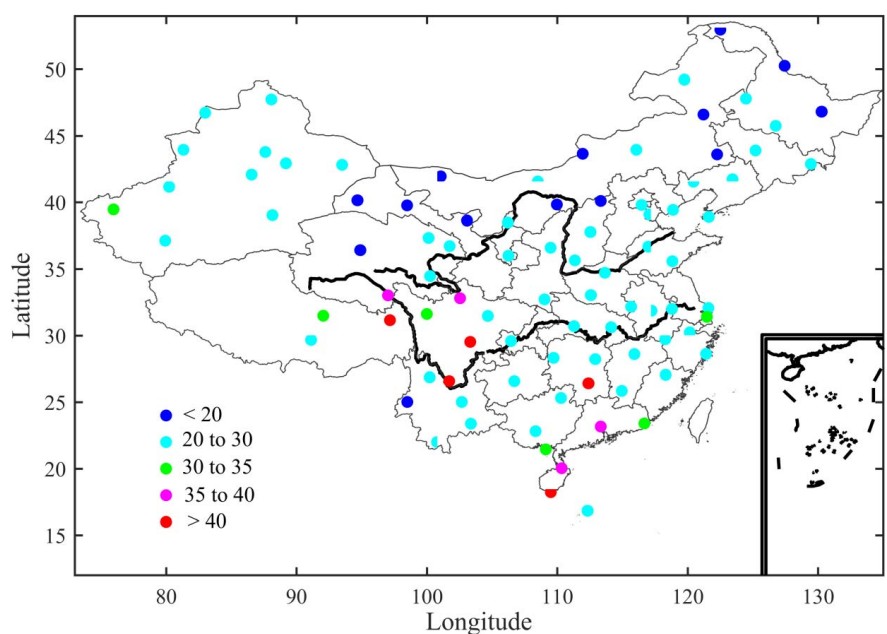


**Figure 8** Spatial distribution of RMSE (W m$^{-2}$) for our estimated daily SSR (spatial

resolution 90 km) at 90 CMA radiation stations.







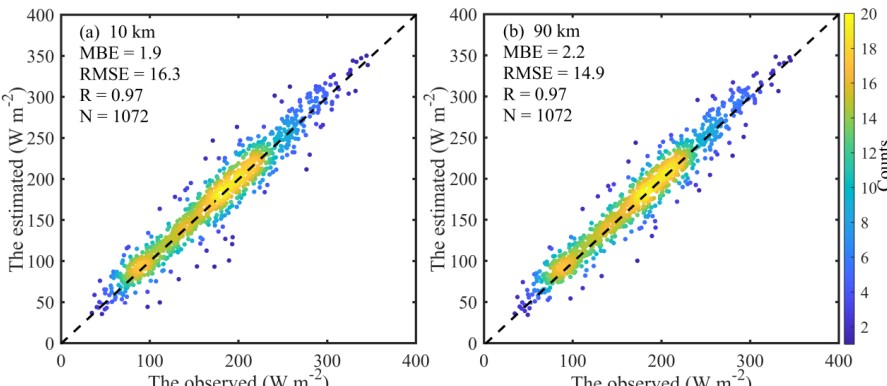


**Figure 9** Comparisons of our estimated monthly SSR at spatial resolutions of (a) 10


km and (b) 90 km with observed monthly SSR at 90 CMA radiation


stations.








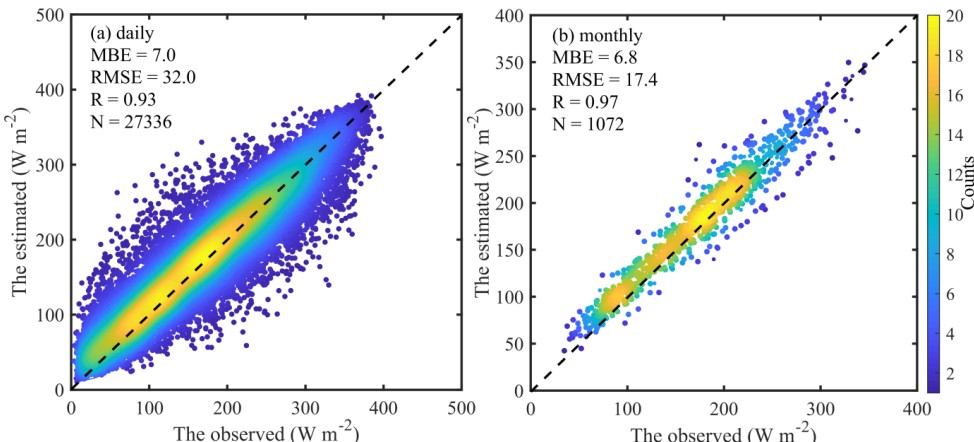


**Figure 10** Comparison of CERES (a) daily and (b) monthly SSR products with those

observed at 90 CMA stations.




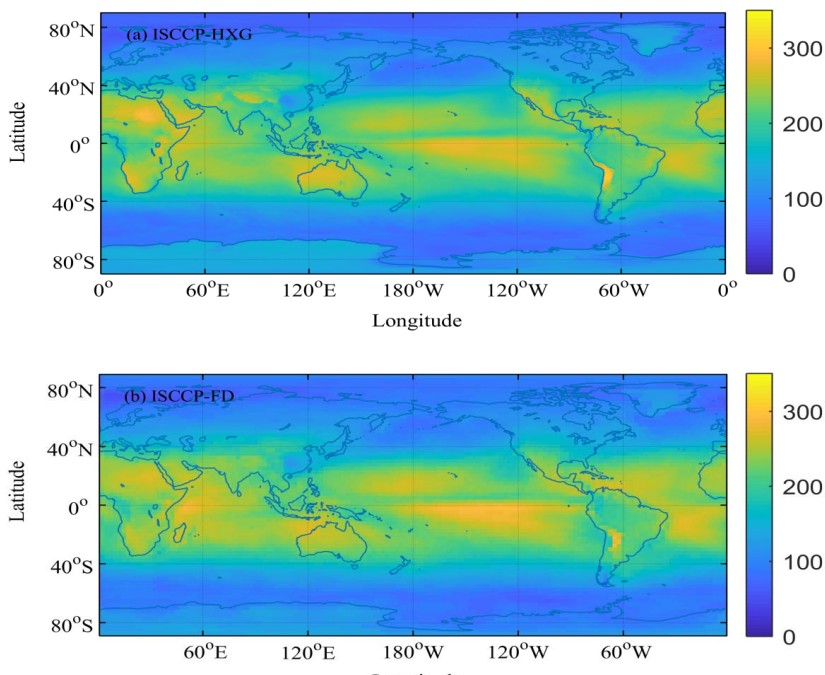


**Figure 11** Spatial distribution of global annual mean SSR (W m$^{-2}$) of (a) ISCCP-HXG

and (b) ISCCP-FD in 2009.







**Table 1**. Effect of spatial resolution on accuracy of our estimated instantaneous SSR

compared to observations at the 42 BSRN stations. A comparisons with

instantaneous SSR of ISCCP-FD is also shown.

|  | Spatial resolution | MBE (W m$^{-2}$) | RMSE (W m$^{-2}$) | $R$ |
|---|---|---|---|---|
| ISCCP-HXG | 10 km | -11.5 | 113.5 | 0.92 |
| ISCCP-HXG | 30 km | -11.0 | 96.5 | 0.94 |
| ISCCP-HXG | 50 km | -11.3 | 93.5 | 0.95 |
| ISCCP-HXG | 70 km | -11.3 | 93.2 | 0.95 |
| ISCCP-HXG | 90 km | -11.1 | 93.4 | 0.95 |
| ISCCP-HXG | 110 km | -11.4 | 94.3 | 0.95 |
| ISCCP-FD | 280 km | -11.2 | 131.4 | 0.89 |






**Table 2**. Effect of spatial resolution on accuracy of our estimated daily SSR compared
to observations at 42 BSRN stations. A comparisons with daily SSR of
ISCCP-FD is also shown.

| | Spatial resolution | MBE (W m$^{-2}$) | RMSE (W m$^{-2}$) | R |
|---|---|---|---|---|
| ISCCP-HXG | 10 km | -6.1 | 38.0 | 0.93 |
| ISCCP-HXG | 30 km | -5.8 | 33.9 | 0.94 |
| ISCCP-HXG | 50 km | -6.0 | 33.4 | 0.95 |
| ISCCP-HXG | 70 km | -5.9 | 33.3 | 0.95 |
| ISCCP-HXG | 90 km | -5.8 | 33.1 | 0.95 |
| ISCCP-HXG | 110 km | -6.0 | 33.4 | 0.95 |
| ISCCP-FD | 280 km | -6.7 | 51.0 | 0.87 |






**Table 3**. Effect of spatial resolution on accuracy of our estimated daily SSR compared
to observations at 90 CMA radiation stations. A comparison with daily SSR
of ISCCP-FD is also shown.

| | Spatial resolution | MBE (W m$^{-2}$) | RMSE (W m$^{-2}$) | R |
|---|---|---|---|---|
| ISCCP-HXG | 10 km | 1.8 | 32.4 | 0.93 |
| ISCCP-HXG | 30 km | 2.1 | 28.5 | 0.95 |
| ISCCP-HXG | 50 km | 2.2 | 27.4 | 0.95 |
| ISCCP-HXG | 70 km | 2.2 | 27.1 | 0.95 |
| ISCCP-HXG | 90 km | 2.1 | 26.9 | 0.95 |
| ISCCP-HXG | 110 km | 2.1 | 26.9 | 0.95 |
| ISCCP-FD | 280 km | -1.2 | 36.5 | 0.91 |






**Table 4**. Effect of spatial resolution on accuracy of our estimated monthly SSR

compared to observations at 90 CMA radiation stations. A comparison with

monthly SSR of ISCCP-FD data is also shown.

| | Spatial resolution | MBE (W m$^{-2}$) | RMSE (W m$^{-2}$) | R |
|---|---|---|---|---|
| ISCCP-HXG | 10 km | 1.9 | 16.3 | 0.97 |
| ISCCP-HXG | 30 km | 2.2 | 15.3 | 0.97 |
| ISCCP-HXG | 50 km | 2.2 | 15.0 | 0.97 |
| ISCCP-HXG | 70 km | 2.2 | 14.9 | 0.97 |
| ISCCP-HXG | 90 km | 2.2 | 14.9 | 0.97 |
| ISCCP-HXG | 110 km | 2.1 | 14.8 | 0.97 |
| ISCCP-FD | 280 km | -1.3 | 20.0 | 0.95 |
