# Peer review of "A 16-year dataset (2000-2015) of high-resolution (3 hour, 10 km) global surface"

_Earth System Science Data, 2019_

## Referee Comment (RC1) · Anonymous Referee #1 · 20 Aug 2019

In this paper, the authors generated 16-year dataset of surface solar radiation (SSR) with high resolution, according to the latest ISCCP and ERA5 data. The SSR is required for land surface process simulations and solar energy estimation. The proposed method for SSR estimation is the physical scheme that used in the previous studies of Qin and Tang et al. The paper is clear and well written. However, the following questions are not satisfactorily answered:

(1). In recent years, Zhang et al 2014 developed SSR high-resolution products based on multi-source satellite data e.g. MODIS. It is recommended that the author describe the characteristics of the study in Line 104, indicating the differences and individual

characteristics of the products developed in this study.

Zhang, Xiaotong, et al. "Generating Global LAnd Surface Satellite incident shortwave radiation and photosynthetically active radiation products from multiple satellite data." Remote Sensing of Environment 152 (2014): 318-332.

(2). L157, cloud top temperature was used to discriminate the water and ice cloud, are there any more details about this? MODIS has a cloud top temperature product. Why not use this product?

(3). L166, MOD08 was used to provide aerosol data, what's the parameters used? AOD?

(4). MOD08 has three temporal resolutions: daily, 8-day and monthly. So which one was used in your SSR calculation, and how do you solve the course temporal resolution of MOD08 to match the high temporal of other input data? Please write the details information on this in your manuscript.

(5). Validated data, 42 stations from BSRN and 90 stations from CMA were used to evaluate the performance of the estimated SSR. For site data applications, have quality control of site data during the verification process of remote sensing products? What standard to control?

---

## Author Comment (AC1) · 25 Sep 2019

**Response to Referee #1**

We would like to thank the reviewer for the comments and suggestions, which help to improve the quality of our work. We have made revisions and have replied to all comments and suggestions. Please find a detailed point-by-point response to each comment.

**Comment:**

In this paper, the authors generated 16-year dataset of surface solar radiation (SSR) with high resolution, according to the latest ISCCP and ERA5 data. The SSR is required for land surface process simulations and solar energy estimation. The proposed method for SSR estimation is the physical scheme that used in the previous studies of Qin and Tang et al. The paper is clear and well written. However, the following questions are not satisfactorily answered:

**Response:**

We thank Referee #1 for the encouraging comments. All comments and suggestions have been considered carefully and well addressed.

**Comment:**

1. In recent years, Zhang et al 2014 developed SSR high-resolution products based on multi-source satellite data e.g. MODIS. It is recommended that the author describe the characteristics of the study in Line 104, indicating the differences and individual characteristics of the products developed in this study.

Zhang, Xiaotong, et al. "Generating Global LAnd Surface Satellite incident shortwave radiation and photosynthetically active radiation products from multiple satellite data." Remote Sensing of Environment 152 (2014): 318-332.

**Response:**

More description about Zhang's article will be added into in the revised manuscript as "The GLASS SSR products were retrieved by a look-up table method with the visible band top-of-atmosphere (TOA) radiance from multi-source geostationary and polar-orbiting satellite data".

**Comment:**

2. L157, cloud top temperature was used to discriminate the water and ice cloud, are there any more details about this? MODIS has a cloud top temperature product. Why not use this product?

**Response:**

More details about the determination of cloud phases will be added in the revised

manuscript as "In the ISCCP H-series cloud product, cloud types are roughly defined by two phases (liquid and ice), which are determined by cloud top temperature (TC) with liquid for TC ≥ 253.1 K, and ice for TC < 253.1 K."

Yes, MODIS has a cloud top temperature product, but there are big mismatches between the times of MODIS and ISCCP H-series cloud product, which would lead to great uncertainty.

**Comment:**
3. L166, MOD08 was used to provide aerosol data, what's the parameters used? AOD?
4. MOD08 has three temporal resolutions: daily, 8-day and monthly. So which one was used in your SSR calculation, and how do you solve the coarse temporal resolution of MOD08 to match the high temporal of other input data? Please write the details information on this in your manuscript.

**Response:**

In this study, we used the MODIS AOD product of the combined dark target and deep blue AOD at 0.55 micron for land and ocean. Thus, the information about the MODIS aerosol will be added in the revised manuscript as "The MODIS AOD product of the combined dark target and deep blue AOD at 0.55 micron for land and ocean was used".

In this study, we used the MOD08 daily product. Thus, the word "MOD08 and MYD08" in the original manuscript will be changed to "MOD08_D3 and MYD08_D3" in the revised manuscript.

To match the temporal of ISCCP HXG products, we re-sampled MODIS aerosols and albedo to 3 hour by assuming that their values are constant within a day. This sentences will be added in the revised manuscript.

**Comment:**
5. Validated data, 42 stations from BSRN and 90 stations from CMA were used to evaluate the performance of the estimated SSR. For site data applications, have quality control of site data during the verification process of remote sensing products? What standard to control?

**Response:**

The BSRN radiation data used in this study were quality controlled by station scientists before release, and are regarded as the most reliable radiation data due to the instruments of highest available accuracy and careful maintenance.

The CMA radiation data used in this study were quality controlled by a two-steps procedure developed by Tang et al. (2010). One is the physical threshold test to eliminate the obvious errors, and the other is the statistical test using artificial neural network method to eliminate the more insidious errors. More detailed information about the two-steps procedure can be found in the article of Tang et al. (2010).

Tang, W., Yang, K., He, J., and Qin, J.: Quality control and estimation of global solar radiation in China, Sol. Energy, 84, 466–475, 2010.

---

## Referee Comment (RC2) · Guanghui Huang (Referee) · 11 Nov 2019

This paper describes a 16-year global Surface Solar Radiation (SSR) dataset that is produced using the newest ISCCP cloud products. The new SSR data is more accurate than ISCCP-FD, CEWEX-SRB, which may provide a better alternative for surface studies of hydrology, ecology and land processes because SSR is a basic input for them. The paper is well written and organized. Therefore, I recommend its publishing on the Earth System Science Data.

Detailed review 1. Line 22 with -> using 2. Line 31-34, please rephrase this sentence. 3. Line 78-79. This sentence seems awkward, please rephrase it. There is a high-level

discussion on the category of how to derive SSR from satellites in the review paper of Huang et al (2019). Huang, G.H., Li, Z.Q., Li, X., Liang, S.L., Yang, K., Wang, D.D., & Zhang, Y., 2019. Estimating surface solar irradiance from satellites: Past, present, and future perspectives. Remote Sensing of Environment, 233. 4. Line 344. I would suggest you delete the word of "retrieval". 5. The first paragraph of Section 6 seems a little bit wordy and boring. Please consider to condense it.

---

## Author Comment (AC2) · 13 Nov 2019

**Response to Referee #2**

We would like to thank the reviewer for the comments and suggestions, which help to improve the quality of our work. We have made revisions and have replied to all comments and suggestions. Please find a detailed point-by-point response to each comment.

**Comment:**

This paper describes a 16-year global Surface Solar Radiation (SSR) dataset that is produced using the newest ISCCP cloud products. The new SSR data is more accurate than ISCCP-FD, CEWEX-SRB, which may provide a better alternative for surface studies of hydrology, ecology and land processes because SSR is a basic input for them. The paper is well written and organized. Therefore, I recommend its publishing on the Earth System Science Data.

**Response:**

We thank Referee #2 for the encouraging comments. All comments and suggestions have been considered carefully and well addressed.

**Comment:**

1. 1. Line 22 with -> using

**Response:**

Accepted!

**Comment:**

2. Line 31-34, please rephrase this sentence.

**Response:**

We have changed the sentence to "When the estimated instantaneous SSR data were upscaled to 90 km, its error was clearly reduced with RMSE decreasing to 93.4 W m$^{-2}$ and R increasing to 0.95." in the revised manuscript.

**Comment:**

3. Line 78-79. This sentence seems awkward, please rephrase it. There is a high-level discussion on the category of how to derive SSR from satellites in the review paper of Huang et al (2019). Huang, G.H., Li, Z.Q., Li, X., Liang, S.L., Yang, K., Wang, D.D., & Zhang, Y., 2019. Estimating surface solar irradiance from satellites: Past, present, and future perspectives. Remote Sensing of Environment, 233.

**Response:**

We have changed the sentence to "The many methods that have been developed to retrieve SSR from satellite data can be roughly divided into two categories: statistical methods and methods based on radiative transfer processes (Huang et al., 2019). According to Sengupta et al. (2018), these methods can also be subdivided into three types: empirical, semi-empirical and physical" in the revised manuscript.

**Comment:**

4. Line 344. I would suggest you delete the word of "retrieval".

**Response:**

Accepted!

**Comment:**

5. The first paragraph of Section 6 seems a little bit wordy and boring. Please consider to condense it.

**Response:**

We have changed the first paragraph of Section 6 to "This study produced a 16-year (2000-2015) global dataset of SSR (with resolutions of 3 h and 10 km) based on recently updated ISCCP H-series cloud products, new ERA5 reanalysis data and MODIS albedo and aerosol products with a physically based scheme. The retrieved SSR dataset was evaluated globally with observations collected at BSRN and CMA radiation stations. Validation against observations collected at BSRN showed that the MBE and RMSE were –11.5 and 113.5 W m$^{-2}$ for the instantaneous SSR estimates, and -6.1 and 38.0 W m$^{-2}$ for the daily SSR estimates, but their accuracies were clearly improved when upscaled to more than 30 km. For example, the RMSEs were decreased to 93.4 and 33.1 W m$^{-2}$ when our estimates were upscaled to 90 km. Validation against observations collected at CMA indicated that our estimates of daily and monthly SSR produced RMSE values of 32.4 and 16.3 W m$^{-2}$, respectively, but these values were decreased to 26.9 and 14.9 W m$^{-2}$ when our estimates were

upscaled to 90 km. Comparisons with other global satellite SSR products indicated that the accuracies of our SSR estimates were clearly higher than those of GEWEX-SRB, ISCCP-FD and CERES products." in the revised manuscript.